# A cosmic-ray loaded nascent outflow driven by a massive star cluster

Marianne Lemoine-Goumard [1,6,7] ✉, Lucia Härer [2,7] ✉, Lars Mohrmann [2,7] ✉, Romain Bernet[3,4], Jim Hinton[2], Giada Peron[5], Brian Reville [2], Luigi Tibaldo [3] & Thibault Vieu[2]

Cosmic rays are widely held to drive outflows from star-forming galaxies and profoundly influence galaxy evolution. Direct evidence for cosmic-ray carrying outflows is however lacking. At the same time there is increasing awareness of the importance of massive star clusters in the acceleration of cosmic rays in galaxies. Here we report on the discovery of a nascent outflow driven by the massive star cluster Westerlund 1. Giga-electronvolt gamma-ray emission coincident with a cavity visible in atomic hydrogen traces the emergence of a population of relativistic electrons out of the Galactic Disc. The emission is offset from tera-electronvolt gamma-ray radiation surrounding the cluster, but connects to it smoothly spectrally and spatially. The implied energy density of co-accelerated protons and nuclei, assuming standard non-thermal electron/proton injection efficiencies, is at least an order of magnitude higher than that in the general interstellar medium. These particles therefore have the potential to dynamically influence the outflow. This discovery suggests that cosmic-ray loaded outflows may be a common feature of young massive star clusters, with implications for the transport of cosmic rays into the halo of the Galaxy.

Observations of nearby starburst galaxies suggest the presence of global cosmic-ray (CR) pressure driven winds[1]. It is unclear however whether the CR content in the halo of the Milky Way is sufficient to accelerate such a wind. On the other hand, galaxy-scale simulations are in agreement that CR feedback at high latitudes succeeds in driving large-scale winds[2-7]. Since these winds affect the nearby circumgalactic medium and, as a consequence, the Galactic star-formation rate, the CR content in the Galactic halo and the processes that regulate how CRs get there are critical components of any model of the interstellar medium (ISM)[8].

Early fluid[9,10] and kinetic[11,12] models predicted the existence of Galactic winds driven by CRs in the halo. Modern simulations—encompassing stellar feedback, multiple supernovae, the resulting CR production and interaction with a multi-phase ISM—can now present a more consistent picture of this process e.g. [2-7]. These works find that the kpc-scale outflows are sensitive not only to the gas conditions in and around superbubbles, but also to the assumptions on CR transport and associated plasma effects. These effects, such as ion-neutral damping, depend upon the phase of the surrounding ISM and in particular that above and below the Galactic Plane.

The material feeding these halo winds originates from stellar activity, especially supernovae[13,14], which are unevenly distributed throughout the disk. To reach the halo, material must penetrate the diffuse warm ISM (WIM; see ref. 15 for further details). The WIM density falls off exponentially away from the mid-plane, with a scale-height of ≈1 kpc[16], substantially longer than that of the thin disc of the Galactic Plane (≈100 pc[14]). Here, young massive star clusters (YMSCs) have a distinct advantage. The combined action of powerful winds and

[1]Université Bordeaux, CNRS, LP2I Bordeaux, UMR 5797, Gradignan, France. [2]Max-Planck-Institut für Kernphysik, Heidelberg, Germany. [3]IRAP, Université de Toulouse, CNRS, CNES, UPS, Toulouse, France. [4]Institut Supérieur de l'Aéronautique et de l'Espace (ISAE-SUPAERO), Université de Toulouse, Toulouse, France. [5]INAF Osservatorio Astrofisico Arcetri, Florence, Italy. [6]Present address: Université Bordeaux, CNRS, LP2I Bordeaux, UMR 5797, Gradignan, France. [7]These authors contributed equally: Marianne Lemoine-Goumard, Lucia Härer, Lars Mohrmann. ✉e-mail: mlemoine33@gmail.com; lucia.haerer@mpi-hd.mpg.de; lars.mohrmann@mpi-hd.mpg.de

clustering of supernovae in the first 10 Myr of the YMSC's evolution are known to inflate superbubbles, that can extend over several hundreds of pc, exceeding the scale-height of the Galactic Plane. Due to the gradient in external gas density normal to the Galactic Disc, the bubbles can become asymmetric and expand preferentially down the gradient forming structures called chimneys[17,18].

The connection between superbubble-driven outflows and CRs is especially significant, given the growing interest in YMSCs as CR sources[19], and in particular their contribution to Galactic CRs at the highest energies[20,21]. Gamma-ray observations of YMSCs at giga-electronvolt (GeV) energies e.g. [22-27] as well as tera-electronvolt (TeV) energies[28-33] indicate they are sites of efficient CR production. Gamma-ray observations can therefore be used as probes of superbubble-driven outflows, constraining the entrained CR content therein.

Westerlund 1, as the most massive YMSC observed in our Galaxy, offers an ideal target for gamma-ray studies. Its cluster core is located at a distance of about 4 kpc[34,35] and is known to host a wealth of massive stars[36,37]. Multiple estimates place the age of the cluster at around 4 Myr[36-38], although recent studies have found evidence for sub-populations of stars with ages of up to 10 Myr[34,39]. Within the last Myr, the kinetic wind-power output of the cluster has plausibly been of the order of $10^{39}$ erg s$^{-1}$[40].

Westerlund 1 is unique among YMSCs in that it is surrounded by a resolved gamma-ray 'ring' at TeV energies, spanning radii of about 20–50 pc[30]. This ring is thought to be connected with the superbubble around the cluster and has been hypothesised to arise from inverse-Compton (IC) emission of high-energy electrons accelerated at the termination shock of the collective cluster wind[41]. CR protons and nuclei ('hadronic CRs'), although almost inevitably produced alongside electrons, produce effectively measurable gamma-ray emission only in the presence of relatively dense target material. The hot, low-density gas expected in an outflow from Westerlund 1 is therefore not favourable for hadronic gamma-ray production. The accelerated electrons, on the other hand, will inevitably cool via inverse-Compton scattering, with a large fraction of the available power converted to gamma rays. As the most massive YMSC observed in the Galaxy, with established ongoing CR acceleration to beyond 100 TeV, Westerlund 1 presents the most promising target to understand the propagation of accelerated CRs away from their sources, and probe the emergence of a large-scale outflow. To probe the system as a whole requires electrons with cooling times comparable to the age of the cluster itself, corresponding to energies of ~30 GeV (see 'Methods') and resulting gamma-ray emission at around 3 GeV, in the range of the *Fermi*-LAT instrument[42].

Here we show that the electron population that produces the TeV gamma-ray emission connects smoothly to the expected large scale outflow emerging below the Galactic Plane. The entrained electrons are revealed by their inverse-Compton emission which we detect with the *Fermi*-LAT instrument. The presence of such an outflow is consistent with a low-density feature seen in relevant gas maps. These findings indicate the presence of a nascent cosmic-ray loaded Galactic outflow.

## Results and discussion
### *Fermi*-LAT data analysis
Using 15 years of *Fermi*-LAT data, we carried out a deep morphological and spectral analysis of the gamma-ray emission in a 15° × 15° region around Westerlund 1 in the 3 GeV–3 TeV energy band. As our aim is to connect with the H.E.S.S. measurements of the Westerlund 1 region, and to reduce uncertainties related to the modelling of diffuse gamma-ray emission, we do not consider gamma rays below 3 GeV in this work. We put special emphasis on accurately modelling the diffuse interstellar radiation arising from interactions of the Galactic CR sea with gas and magnetic fields (see 'Methods').

The result of the *Fermi*-LAT analysis is illustrated in Fig. 1. Our study reveals two new components with respect to previous studies: (i)

an emission region coincident with Westerlund 1 that we model with a template derived from the H.E.S.S. TeV flux map[30]; (ii) a diffuse component extending from the cluster location away from the Galactic Plane, best represented by a Gaussian source ($l = (339.61 \pm 0.03)°$, $b = (-1.91 \pm 0.04)°$, $\sigma = (0.71 \pm 0.03)°$; shown by the dashed orange circle in Fig. 1). We will refer to this second new component as J1654−467.

As we demonstrate in the 'Methods' section, the first component represents a counterpart to the ring-like TeV gamma-ray emission detected with H.E.S.S., being compatible both in terms of spatial morphology and energy spectra. J1654−467, on the other hand, appears to be a continuation of the TeV gamma-ray radiation detected with H.E.S.S., which features emission protruding from the ring in the same direction (named HESS J1652−462 in ref. 30). The structure is ~150 pc long in projection and, with a gamma-ray energy flux above 10 GeV of $(10.5 \pm 0.5_{\text{stat}} \pm 0.4_{\text{syst}}) \times 10^{-11}$ erg cm$^{-2}$ s$^{-1}$, is the dominant source of 10–100 GeV photons in the region. That is in contrast to the picture at TeV energies, where the 'annex' is less extended and not as bright as the emission surrounding the star cluster. This is naturally explained by the energy-dependent cooling rate of radiating electrons (see 'Methods'). In contrast, the indication of spectral softening with distance from Westerlund 1 that we observe from the *Fermi*-LAT data themselves (see 'Methods') is not explained by the difference in cooling time scales, but could be a feature of energy-dependent particle transport. A search for possible multi-wavelength counterparts to J1654−467 did not reveal any candidates besides Westerlund 1. In particular, we find that the two pulsars PSR J1648−4611 and PSR J1650−4601, whose positions on the sky are coincident with the southern part of the 'TeV ring'−while making a dominant contribution to the emission below 10 GeV−cannot be invoked to explain the presence of J1654−467 (see 'Methods').

### Interstellar medium density
A natural explanation for the new structure is that it traces a population of relativistic particles accelerated at or near the cluster Westerlund 1 that is breaking through the Galactic Plane in a nascent outflow (see Fig. 2). Such outflows are expected for superbubble expansion in a stratified medium[15,43].

This scenario can be tested by looking for an under-density, or cavity, in the Galactic gas emission that such a nascent outflow would produce. Because no dense molecular hydrogen gas is present near J1654−467 (see Supplementary Fig. 3), we focus here on atomic hydrogen gas. Figure 3 shows H I emission at velocities (with respect to the local standard of rest) around the Galactic rotation velocity of Westerlund 1 ($v_{\text{LSR}} \approx -50$ km s$^{-1}$, see 'Methods'). A clear under-density is apparent over an extended velocity range, coincident with the peak *Fermi*-LAT emission. This under-density manifests itself in a difference in column density of ~$(0.7–1.5) \times 10^{20}$ cm$^{-2}$ with respect to neighbouring lines of sight. Assuming that the depth of the structure is similar to its width, this corresponds to a deficit of about 0.3–0.7 hydrogen atoms cm$^{-3}$ (see 'Methods'). The energy required to inflate this structure depends on the assumed external temperature ($T$), the depth in projection ($d$) and the nature of the contents, but is of order $10^{50}$ ($T$/$10^4$ K)($n$/1 cm$^{-3}$)($d$/100 pc)$^3$ erg, which is readily available given the power of Westerlund 1.

### Modelling
The spectral energy distribution (SED), as presented in Fig. 4, demonstrates that the total *Fermi*-LAT spectrum connects smoothly to the H.E.S.S. observed TeV gamma-ray emission surrounding Westerlund 1. The same is found to hold true in different sub-regions (see 'Methods'). This suggest a common origin of the emission from the entire region. As discussed above, a leptonic interpretation presents a natural solution to both the lack of correlation with gas that one would expect in a hadronic dominated scenario, and the energy-dependent morphology. This interpretation complements the association of the

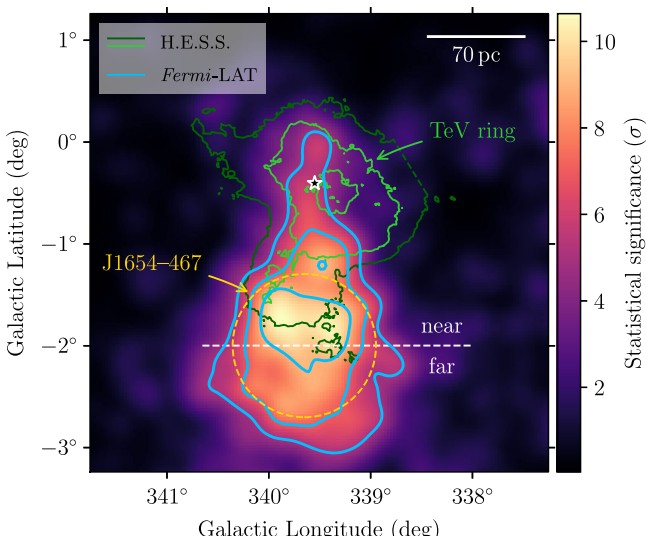

**Fig. 1 | *Fermi*-LAT test statistic (TS) map.** The colour scale displays $\sqrt{TS}$, i.e. the statistical significance of a point-like source with a spectrum $\propto E^{-2}$, in the energy range 3 GeV–3 TeV. Contour lines at $\sqrt{TS} = (5, 7, 9)$ are shown in blue. The orange, dashed circle denotes the 1-$\sigma$ radius of the Gaussian source model for J1654−467. Flux contours of the TeV gamma-ray emission measured with H.E.S.S.[30] at levels of $(1.1/2.3) \times 10^{-8}$ cm$^{-2}$ s$^{-1}$ sr$^{-1}$ are shown in dark and light green, respectively. The star marker indicates the position of Westerlund 1. The white, dashed line separates the J1654−467 region into two halves, for which we derive separate spectra (cf. Fig. 4). The scale bar assumes a distance to Westerlund 1 of 4.14 kpc[34,35].

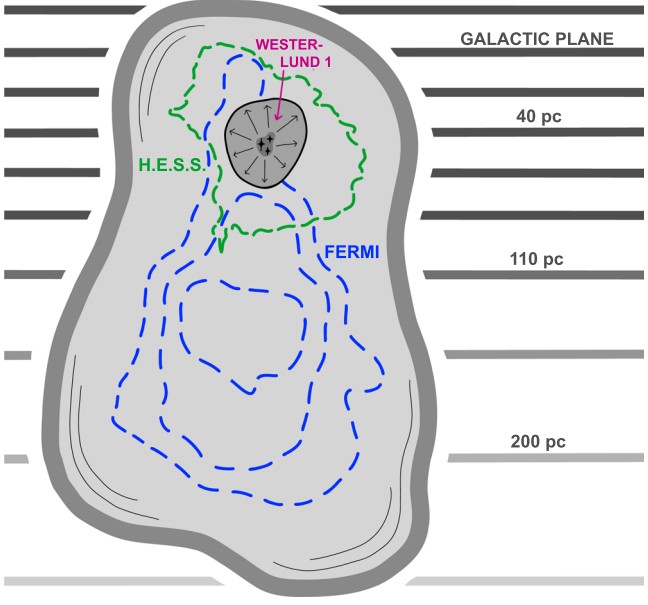

**Fig. 2 | Sketch of nascent outflow around Westerlund 1.** Massive stars and supernovae in Westerlund 1 excavate material around the cluster, resulting in a low-density bubble, which expands asymmetrically due to the density gradient in the Galactic Disc. This will eventually lead to the formation of an open outflow connecting to the halo. Cosmic rays are accelerated at the cluster wind termination shock (black line). High-energy electrons exhibit short cooling times and generate the TeV-energy gamma rays measured with H.E.S.S. (green contour[30]); lower-energy electrons can travel further and are transported along the nascent outflow, where they are responsible for producing the GeV gamma-ray emission detected with *Fermi*-LAT (blue contours). Note that distances are not corrected for projection effects, i.e. the same scale as in Fig. 1 is used.

TeV emission with electrons accelerated at the termination shock of the star cluster wind of Westerlund 1[30,41]. We propose that the gamma rays in J1654−467 originate from electrons accelerated at the wind termination shock which accumulate in the nascent outflow. In this scenario, the Westerlund 1 superbubble expands asymmetrically due to the density gradient normal to the plane of the disk, as illustrated in Fig. 2. The shock-heated material in the bubble's interior thus preferentially flows away from the Galactic mid-plane, entraining the large-scale magnetic fields. Consequently, the accelerated particles will follow the bulk flow, subject to advection and diffusive transport.

A leptonic inverse-Compton model for the SED of the entire region is presented, extending the model of ref. 41 to GeV energies. We refer to this as the *total model*. Figure 4 shows the model curve for an electron injection index of 2.25, a cluster age of 4 Myr, wind power of $10^{39}$ erg s$^{-1}$ and a constant magnetic field of 2 μG. Note that due to the illustrative nature of the model, these values are merely representative of the average behaviour and might be revised in future, more detailed modelling. The required efficiency for conversion of the wind's kinetic energy to accelerated electrons at the cluster's wind termination shock is 0.7% above 0.01 GeV, which lies in the typical expected range. In constructing the model, the software package GAMERA[44,45] is used to model the evolution of a continuously injected particle spectrum, and the resulting gamma-ray emission. We take into account inverse-Compton scattering on the CMB, diffuse Galactic radiation fields, and starlight from Westerlund 1 (see 'Methods'). The contribution of synchrotron radiation to the spectrum in the energy range shown in Fig. 4 is negligible. In the radio domain, the level of synchrotron emission predicted by the model is well below upper limits derived from radio continuum maps (see Supplementary Fig. 4).

To further investigate the scenario of CR transport along the nascent outflow, we present a second model to fit only the emission of the 'far' outflow region as defined in Fig. 1 (*far outflow model*). We assume the same energy injection as that of the *total model*, but fix the period during which injection occurred to match the normalisation in the far region. The required time-scale of injection is 0.5–1 Myr. The high-energy cut-off in the far outflow spectrum can be reproduced if no new particles were injected into this region in the last ~125–200 kyr, corresponding to the cooling time of $\lesssim$ TeV energy electrons (see 'Methods'). This can be interpreted as the transport time-scale from the acceleration site (the cluster wind termination shock of Westerlund 1) to the far outflow.

The time-scale for transport to the far outflow is much shorter than expected for advection in a spherically symmetric superbubble, indicating a substantial deviation of the flow profile from $\propto 1/R^2$, or severe impact of diffusion. Diffusion time-scales for various diffusion coefficients are given in the 'Methods' section. The observations presented here call for detailed multi-dimensional modelling to connect between different regions and to better constrain the transport and confinement of particles within the nascent outflow. Ultimately, this will elucidate transport properties of CRs in superbubble environments and clarify the role of star clusters in the Galactic CR ecosystem. Using the required injection efficiency of 0.7%, one can estimate the total energy density of relativistic electrons in the nascent outflow to be $U_e \sim 1$–10 eV cm$^{-3}$. Diffusive shock acceleration theory predicts that relativistic protons carry a larger fraction of the total energy flux, and though not measurable in the low density cavity, would carry a substantially larger energy density. This equates to an energy density orders of magnitude greater than the locally measured CR spectrum. This highlights the possibility that CRs can be dynamically important in regions shaped by stellar feedback.

## Concluding remarks

The newly discovered gamma-ray emission, connecting the most massive young star cluster in the Galaxy to a nascent outflow, represents a major step forward for our understanding of the role of young star clusters in CR acceleration, and of CR transport.

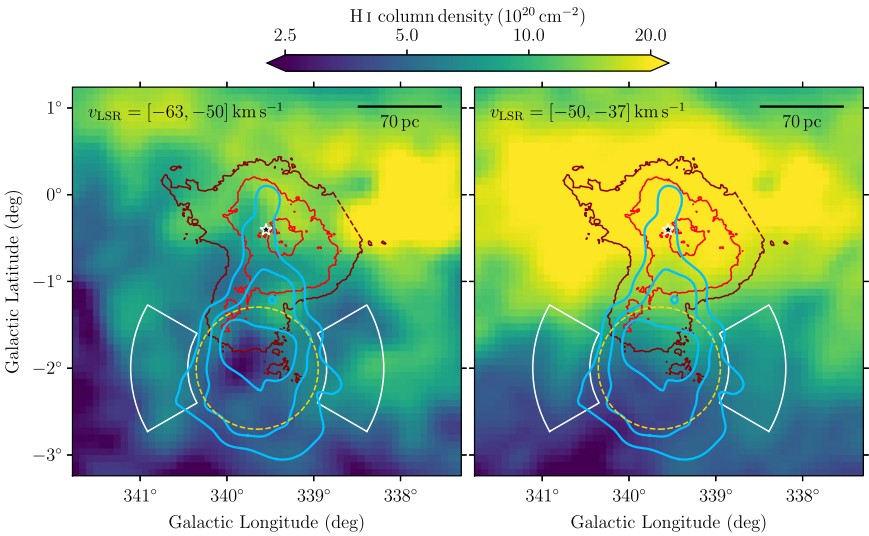

**Fig. 3 | Atomic (H ɪ) gas maps of the region around Westerlund 1.** Data are taken from the Parkes Galactic All-Sky Survey third data release (GASS III)[62]. The maps are derived using the optically thin approximation and are shown for the velocity ranges indicated in each panel. *Fermi*-LAT GeV emission contours (blue), extent of J1654−467 (dashed orange), H.E.S.S. TeV flux contours (here red) and position of Westerlund 1 (star marker) are the same as in Fig. 1. The white arc-shaped regions have been used to compute the difference in gas density between the outflow region and its surroundings.

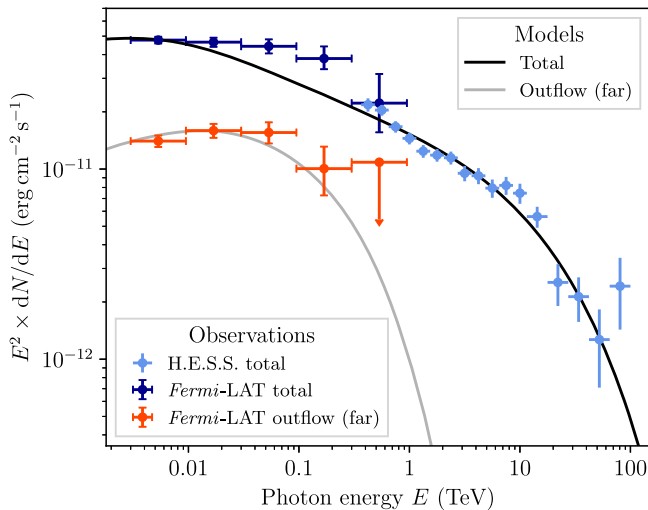

**Fig. 4 | Spectral energy distribution of the gamma-ray emission.** The dark and light blue points show the total flux measured with *Fermi*-LAT (from the 'TeV ring' and J1654−467) and H.E.S.S., respectively, illustrating the spectral continuity despite the visually different spatial distributions. The black line shows a single-zone IC model for the whole population of electrons accelerated in the superbubble. The red points show the spectrum measured in the 'far' region (cf. Fig. 1). Error bars denote 68% c.l. statistical uncertainties; upper limits are at 95% c.l. The grey line follows from a simple time-dependent model for electrons in this region after 125 kyr.

The new *Fermi*-LAT source J1654−467 is consistent with the young superbubble associated to Westerlund 1 being in the early stages of breaking through the edge of the Galactic Plane, and is transporting a significant population of CRs out of the disk. The estimated CR energy density is more than an order of magnitude beyond that of the general ISM; thus CRs might dynamically influence the nascent outflow. The observed situation is dramatically different from the typical assumption of isotropic diffusion of CRs from sources inside the disc. It can be expected that the nascent outflow will eventually provide a channel for CRs accelerated near Westerlund 1 to be transported outside of the disk into the Galactic halo.

Deeper multi-wavelength observations of the Westerlund 1 system are needed to understand the evolution of the nascent outflow and hence the transport of particles into the halo. Structures of this kind may be a common feature of massive star clusters, and gamma-ray (and multi-wavelength) searches around other systems are strongly motivated.

## Methods
### *Fermi*-LAT data analysis
The analysis shown in this paper uses 15 years of *Fermi*-LAT[42] data from 2008 August 4 (MJD 54682) to 2023 August 3 (MJD 60159). Time intervals during which the satellite passed through the South Atlantic Anomaly are excluded. Our data are also filtered removing time intervals around solar flares and bright GRBs, following the procedure used in all *Fermi*-LAT catalogues. The current version of the LAT data is P8R3. The event selection is based on the low-background SOURCE-VETO class with the corresponding instrument response functions (IRFs) P8R3_SOURCEVETO_V3. To reduce contamination from the bright gamma-ray emission from the Earth atmosphere, we select events with zenith angles smaller than 105°. The data reduction and exposure calculations are performed using *fermitools* version 2.2.0 and *fermipy* version 1.2.0. We perform a binned likelihood analysis combining all event types between 3 GeV and 3 TeV with 10 energy bins per decade. We analyse a region of 15° × 15°, centred on Westerlund 1, with spatial bins of 0.03°, including all sources from the *Fermi*-LAT 14-year Source Catalogue (4FGL-DR4; https://fermi.gsfc.nasa.gov/ssc/data/access/lat/14yr_catalog)[46–48] in a region of 25° × 25°.

Interstellar emission contributes substantially to LAT observations in the Galactic Plane, where Westerlund 1 is located. For our study of this complex region, we modelled the spatial and spectral distributions of the interstellar radiation with a linear combination of galactocentric templates ('rings') representing the dark neutral medium (DNM), H ɪ, CO (i.e. gamma-ray emission from hadronic cascades produced in interactions of CRs with DNM, atomic and molecular hydrogen gas), and inverse-Compton components of the emission. These templates are those being used to derive the standard Galactic diffuse background model (gll_iem_v07.fits), but by employing separate scaling factors for the different H ɪ and CO rings (while using an overall normalisation for the inverse-Compton components as well

**Table 1 | Overview of models fitted to the *Fermi*-LAT data**

| ID | Westerlund 1 ring | Outflow | LLH | d.o.f | ΔAIC |
|---|---|---|---|---|---|
| (1) | 3 PS | 2 disks + 3 PS | −840230.4 | 35 | 0 |
| (2) | 3 PS | 1 Gauss | −840183.3 | 18 | 128.2 |
| (3) | H.E.S.S. temp. + 3 PS | 1 Gauss | −840120.5 | 20 | 249.8 |
| (4) | H.E.S.S. temp. + 1 PS | 1 Gauss | −840121.8 | 12 | 263.2 |
| (5) | H.E.S.S. temp. | 1 Gauss | −840128.5 | 8 | 257.8 |
| (6) | H.E.S.S. temp. (split) | 1 Gauss | −840103.8 | 10 | 303.2 |
| (7) | H.E.S.S. temp. (split) + 1 PS | 1 Gauss | −840103.7 | 14 | 295.4 |
| (8) | H.E.S.S. temp. (split) | 1 Gauss (split) | −840098.2 | 13 | 310.6 |
| (9) | 1 disk | 1 Gauss | −840115.1 | 11 | 278.6 |
| (10) | 1 disk (split) | 1 Gauss | −840111.3 | 13 | 282.2 |

Results of the fit of the *Fermi*-LAT data between 3 GeV and 3 TeV using different spatial models. 'Westerlund 1 ring' refers to the region surrounding the star cluster in which TeV emission with a ring-like structure has been observed with H.E.S.S.; 'Outflow' denotes the putative outflow region. 'PS' is for point source. The fourth column reports the log-likelihood (LLH) values obtained for each spatial model, while column 5 indicates the number of degrees of freedom adjusted in the model. The delta Akaike criterion, defined as $\Delta AIC = AIC_1 - AIC_i = 2 \times (\Delta \, d.o.f - \Delta \ln \mathcal{L})$, is reported in the last column.

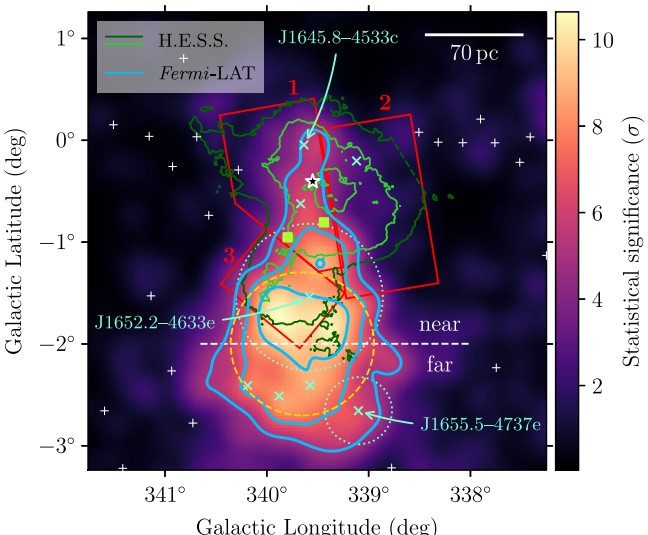

**Fig. 5 | Expanded *Fermi*-LAT TS map.** Same as Fig. 1, with additional information: White plus markers indicate positions of 4FGL-DR4 sources that are part of the fitted ROI model; blue-green × markers of those removed from the model. The dotted circles denote the extension of the disk sources 4FGL J1652.2−4633e and 4FGL J1655.5−4737e. The two green square markers show the position of 4FGL J1648.4−4611 and 4FGL J1650.3−4600 (associated with the pulsars PSR J1648−4611 and PSR J1650−4601, respectively). Shown in red are regions 1, 2, 3, for which separate energy spectra have been derived from *Fermi*-LAT and H.E.S.S. data (cf. Fig. 7).

as one normalisation for the positive component of the DNM), we allowed for many more degrees of freedom in fitting the diffuse emission in our region. To properly extract the gamma-ray flux from the extended sources in our region, the patch component (based on the residual gamma-ray intensity from fitting the overall interstellar emission model) was not included. The residual instrumental background and extragalactic radiation are described by a single isotropic component with spectral shape taken from the tabulated model `iso_P8R3_SOURCEVETO_V3_v1.txt`, as is standard. For further details on the *Fermi*-LAT data analysis, we refer to the Supplementary Material.

Our analysis starts from the baseline model provided by the 4FGL-DR4 catalogue. The first steps of the analysis consist of a re-optimisation of the model in which we re-fit the parameters of the background models and of the brightest sources in the region of interest (ROI). In this procedure, we also test for the presence of additional sources that are not included in the catalogue model but statistically significant in our analysis. This is done by means of a likelihood ratio test, where the test statistic (TS) is defined as $TS = 2(\ln \mathcal{L}_1 - \ln \mathcal{L}_0)$, with $\mathcal{L}_0$ and $\mathcal{L}_1$ the likelihoods of the null hypothesis (background only) and the hypothesis being tested (source plus background), respectively. We iteratively add 6 point-like sources with TS > 25 to the ROI model; all are located more than two degrees away from Westerlund 1. Their positions and TS values are included in Supplementary Table 1.

We then perform the morphological analysis of the gamma-ray emission coincident with Westerlund 1. Since we cannot use the likelihood ratio test to compare models that are not nested, we use the Akaike Information Criterion (AIC;[49]). We calculate $\Delta AIC = AIC_1 - AIC_i = 2 \times (\Delta \, d.o.f \cdot \Delta \ln \mathcal{L})$ to compare the different models with respect to our baseline model (Model 1 in Table 1). Since the preferred model is the one that minimises AIC, here a larger value of ΔAIC corresponds to a more favourable model.

We show in Fig. 5 a map with the positions of all 4FGL-DR4 sources in the region indicated (note that, as we will explain below, the sources marked with blue-green × markers are not part of our final model). Several sources are located within the extent of the TeV emission detected with H.E.S.S.: the two pulsars PSR J1648−4611 and PSR J1650−4601 (associated with 4FGL J1648.4−4611 and 4FGL J1650.3−4600, respectively), three point sources 4FGL J1644.5−4602c, 4FGL J1645.8−4533c, 4FGL J1648.4−4554 and the unidentified extended source 4FGL J1652.2−4633e.

As can be seen from Fig. 6a, b, the two pulsar-associated sources 4FGL J1648.4−4611 and 4FGL J1650.3−4600 dominate the emission below 10 GeV; we therefore include them in all models described in what follows. We note, however, that the two associated pulsars have spin-down luminosities of $2$–$3 \times 10^{35}$ erg s$^{-1}$[50] and thus cannot be invoked to explain the extended emission present to their south, which requires a population of electrons with total power in excess of $10^{36}$ erg s$^{-1}$ (see 'Modelling details'). The centre of the extended source 4FGL J1652.2−4633e lies outside the ring-like structure observed at TeV energies, but with its disk radius of 0.72°[51] the source extends up to the star cluster. With a TS value above 3 GeV of 770, it is very bright and requires a careful analysis. We therefore start the morphological analysis by investigating 4FGL J1652.2−4633e, keeping a log-parabola model for the spectral shape. Interestingly, its southern edge is coincident with three unassociated point sources flagged for confusion effects (see ref. 46), 4FGL J1656.1−4706c, 4FGL J1657.7−4656c and 4FGL J1658.3−4637c, and another unidentified extended source 4FGL

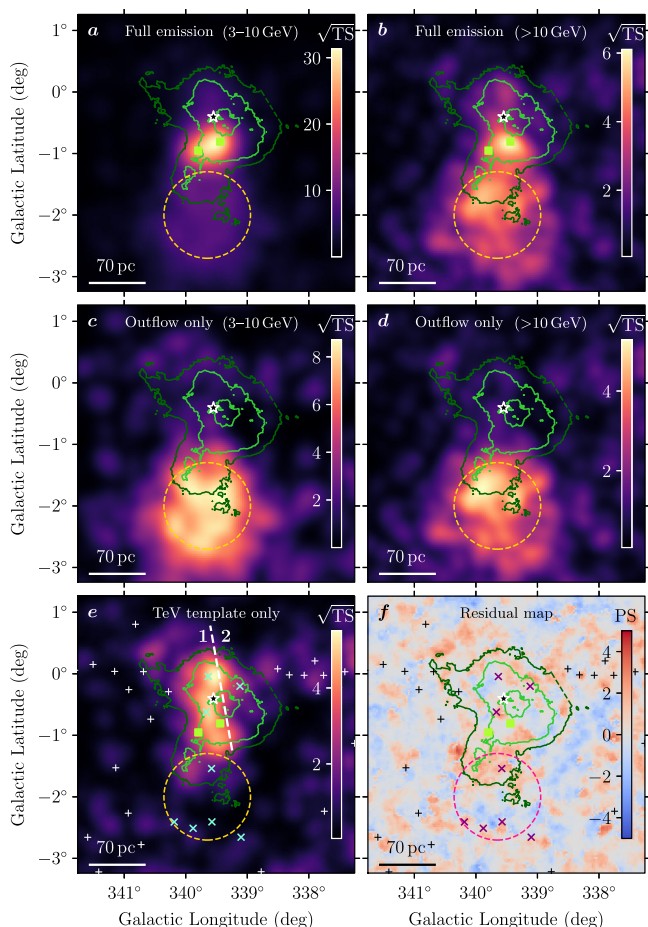

**Fig. 6 | Additional maps from the *Fermi*-LAT data analysis. a–e** show TS maps (as in Fig. 5), highlighting different parts of the emission. **a, b** full emission, including that from the two pulsar-associated sources 4FGL J1648.4–4611 and 4FGL J1650.3–4600 (green square markers), in the energy range 3–10 GeV (**a**) and >10 GeV (**b**). **c, d** emission associated with J1654–467, in the energy range 3–10 GeV (**c**) and >10 GeV (**d**). **e** emission surrounding Westerlund 1, associated with the template derived from the TeV emission measured with H.E.S.S. The white dashed line separates the template into two regions for which separate spectra are derived. **f** PS map[53] for the best-fit model including all components. Contour lines and source markers in all panels are the same as those described in Figs. 1, 5.

the TeV emission detected by H.E.S.S., but brighter in region 1 (i.e. left of the dashed white line) than in region 2. We therefore split the H.E.S.S. template along the division between regions 1 and 2 and fit them separately, thus providing an improved likelihood value (Model 6). We verify that with a split template, there is no need for the additional point source coincident with 4FGL J1645.8–4533c any more (Model 7). To test the scenario in which J1654–467 is produced by energy-dependent cooling of electrons, we also divide the Gaussian model in two parts, parallel to the Galactic Plane (along the near/far division indicated in Fig. 5), providing our best model of the region (Model 8) following the Akaike criterium. Model 8 provides a 2.5$\sigma$ improvement (ΔTS of 11.2 for 3 additional degrees of freedom) with respect to Model 6. The PS map (see Fig. 6f), constructed to display both positive and negative residuals with respect to the final model[53], demonstrates that this model leads to no statistically significant residuals with respect to the observed data. Dividing the Gaussian source into smaller slices does not provide any significant improvement in the fit. Finally, as an additional cross-check we use a simple disk model to fit the gamma-ray emission associated with Westerlund 1 (Model 9), as well as a disk divided into 2 regions (Model 10). While the disk model yields a reasonable fit, the best model remains the H.E.S.S. template divided in 2 parts (Model 8).

We test a simple power-law model and a power law with an exponential cut-off for each component of our best spatial model (i.e. the two halves of the H.E.S.S. template for the region around Westerlund 1 and the two halves of the Gaussian model for the outflow region). The results from these fits are provided in Supplementary Table 2, and corresponding SEDs including systematic uncertainties are displayed in Supplementary Fig. 2.

Concerning J1654–467, we find that the spectrum in the near part extends to higher energies, while there is an indication of a cut-off to the spectrum in the far part. This is in line with our observation that the spatial modelling is improved when splitting J1654–467 into a near and far part. The maps in Fig. 6c, d, which show the spatial morphology of J1654 – 467 in two energy ranges, further support this: above 10 GeV, the far part is less bright than the near one, which is not the case at lower energies. A statistical test yields a significance of around 2$\sigma$ for this difference. Thus, we find an indication of a spectral softening of the emission along the putative outflow, although we cannot claim this with high significance.

The apparent asymmetry in the spectra derived for Westerlund 1 in regions 1 and 2 could originate from differences in how electrons are injected in both regions. That region 2 is equally bright as region 1 in the TeV domain but dimmer in the GeV domain would then imply that only freshly injected electrons are visible in this region. Alternatively, a deviation in spectral index was detected by[30] in a region close to LMXB 4U 1642–45, which is located in region 1 (see also the recent investigation of this region with eROSITA[54]). The region also coincides with the soft and confused 4FGL source J1645.8–4533c. The current statistics does not allow the detection of an additional point source within region 1 (see Table 1), but an unidentified, soft-spectrum source could contaminate the spectrum of Westerlund 1 (region 1) at low energy, thus explaining the asymmetry between regions 1 and 2.

### Comparison between *Fermi*-LAT and H.E.S.S. SEDs

In addition to the comparison of the total gamma-ray emission measured with *Fermi*-LAT and H.E.S.S. shown in Fig. 4, we have also compared spectra for different spatial regions (labelled 1, 2, 3 in Fig. 5). This comparison is shown in Fig. 7, which demonstrates that the spectra in all regions connect smoothly between the energy ranges covered by the two instruments.

*Fermi*-LAT spectra for regions 1 and 2 are a direct result of the modelling described in the previous subsection, where Models 6–8 utilise a H.E.S.S. flux map template divided along the separating line between the two regions. The SED for region 3 has been computed by

J1655.5–4737e represented by a radial disk (cf. Fig. 5). By replacing these 5 model components with a simple Gaussian (Model 2 in Table 1), the likelihood of the fit significantly improves while the number of degrees of freedom is reduced with respect to the baseline model (the large number of degrees of freedom for Model 1 comes about because we re-fit the positions of all sources). This Gaussian component is referred to with J1654 – 467 in this work. Its best-fit Galactic longitude and latitude are $l = (339.61 \pm 0.03)°$ and $b = (−1.91 \pm 0.04)°$, respectively, while the best-fit 1-$\sigma$ radius is $(0.71 \pm 0.03)°$. We note that diffuse gamma-ray emission with a somewhat smaller extent, coincident with the TeV emission protruding from the 'ring' measured with H.E.S.S., had already been reported in ref. 52.

Moving to the vicinity of the cluster, the next step is to use the H.E.S.S. flux map[30] as a template to fit the *Fermi*-LAT data, along with the three unidentified point sources located within the TeV ring structure (Model 3). We find that keeping only one of the three sources overlapping with the H.E.S.S. template, coincident with 4FGL J1645.8–4533c, is sufficient (but required) to provide smooth residuals (Model 4/Model 5). Figure 6e illustrates that the GeV emission surrounding the cluster is generally compatible in spatial structure with

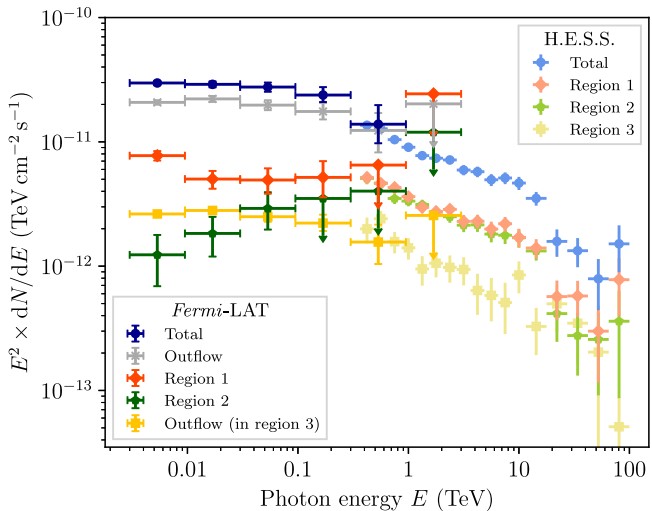

**Fig. 7 | Comparison of energy spectra derived from *Fermi*-LAT and H.E.S.S. data.** Region labels refer to regions as shown in Fig. 5. Error bars denote 68% c.l. statistical uncertainties; upper limits are at 95% c.l.

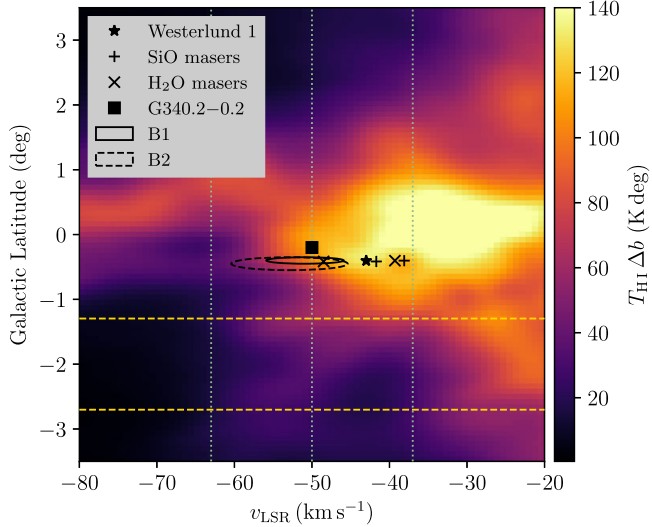

**Fig. 8 | Velocity-latitude diagram of H I emission in the region around Westerlund 1 and J1654−467.** The H I brightness temperatures[62] are integrated over the 1σ longitude range of J1654−467. We show the position of Westerlund 1 member stars[35], SiO and $H_2O$ masers W 26 and W 237[59], the H II region G340.2−0.2[58] and the H I bubble-like features B1 and B2[60]. Dashed horizontal lines show the 1σ latitude range of J1654−467. Dotted vertical lines show the boundaries of the velocity integration ranges used for column-density maps in our study.

integrating the Gaussian spatial model in this region and scaling the total SED by the ratio of that integral to the full model.

To derive the H.E.S.S. SEDs, we sum up the spectra of the square regions a–p defined in ref. 30, taking into account the overlap between the square regions and the regions 1–3 defined here. This assumes that the gamma-ray flux is isotropic within each square region, which is the case in good approximation.

### Galactic kinematics in the direction of Westerlund 1
In order to study the correlation of gamma-ray emission in the J1654−467 region with ISM structures we use emission lines from H I and CO. Based on the plausible physical connection between J1654−467 and Westerlund 1 suggested by the gamma-ray morphology and our modelling work, we aim at studying the ISM in the vicinity of

Westerlund 1. Therefore, we need to identify the range in Doppler-shift velocity with respect to the local standard of rest ($v_{LSR}$) of the lines corresponding to the region of Westerlund 1. It is well known that the kinematics of gas and stars in the direction of Westerlund 1 do not follow closely large-scale rotation curves[35,55]. If we convert the measured distance of $d = 4.14$ kpc (i.e. the average of the estimates presented in refs. 34,35) to a velocity, we obtain $v_{LSR} ≈ -61.7$ km s$^{-1}$, where we have used the rotation curve model from[56] with an orbital velocity of the Sun $V_0 = 220$ km s$^{-1}$ and a galactocentric distance of the Sun $R_0 = 8.178$ kpc[57]. This is inconsistent with the velocity measured from stars in the cluster itself, which is around $v_{LSR} ≈ -43$ km s$^{-1}$[35]. Other kinematic tracers like IR masers and H II regions potentially associated with Westerlund 1 also display velocities in the range from $v_{LSR} ≈ -50$ km s$^{-1}$ to $v_{LSR} ≈ -38$ km s$^{-1}$[58,59].

A dedicated analysis of H I properties in the environment of Westerlund 1[60] led to the identification of two bubble-like features, dubbed B1 and B2, spatially associated with Westerlund 1 at radial velocities ≈ −55 km s$^{-1}$, that is $v_{LSR} ≈ -51$ km s$^{-1}$ and expanding with a velocity of ≈5 km s$^{-1}$. These bubble-like features are interpreted as cavities in the ISM carved by activity of Westerlund 1 itself. In the same velocity range, a third bubble, B3, extends to lower Galactic latitudes. To the north, B3 consists of a shell with two large, bright and complex emission regions to the east and west, while it is open to the south, away from the Galactic Plane[60] in the direction of J1654−467. The same study shows the complexity of H I kinematic structures in this region severely affected by the near-far ambiguity. In the kinematic range discussed above H I emission appears to arise from the superposition of the Norma and Scutum-Crux arms, with Westerlund 1 probably located on the near side of the Norma arm based on recent studies of spiral tracers[61] and the distance estimate from cluster member parallaxes[35].

In this work we use H I data from the Parkes Galactic All-Sky Survey third data release (GASS III)[62] as other higher-resolution surveys do not sufficiently cover the region of J1654−467. Figure 8 shows a velocity-latitude diagram of H I emission in a large region around Westerlund 1 and J1654−467. For the entire velocity range spanned by kinematic tracers associated with Westerlund 1 and by the bubble-like features B1 and B2 from[60] the 1σ latitude range of J1654−467 corresponds to low-emission regions just below the bright emission band from the Galactic Plane.

In acknowledgement of the substantial uncertainties related to the identification of gas in the vicinity of Westerlund 1 and J1654−467 we present gas maps throughout the paper in two velocity ranges, namely $v_{LSR} = [-63, -50]$ km s$^{-1}$ and $v_{LSR} = [-50, -37]$ km s$^{-1}$. These ranges are also indicated in Fig. 8.

### H I column densities in the J1654−467 region
Employing again the GASS III data, we provide in Fig. 3 maps displaying the column density of atomic hydrogen in our region of interest. To convert from brightness temperature to column density, we assume that the gas is optically thin and employ a conversion factor of $X_{HI} = 1.823 × 10^{18}$ cm$^{-2}$ / (K km s$^{-1}$). We verified that, at the high latitude and modest total column density at the location of J1654−467, qualitatively and quantitatively compatible maps are obtained when using a low spin temperature of 100 K.

An under-density with respect to neighbouring lines of sight is visible at the location of J1654−467 in both velocity ranges displayed in Fig. 3. In particular, a clear minimum in density that coincides with the peak of GeV gamma-ray emission can be identified in the left panel. In order to quantify this observation, we compute a rough estimate of the difference in density between the outflow region and its surroundings. We take the region indicated by the yellow dashed circle in Fig. 3 (which corresponds to the 1-σ radius of the Gaussian model used to describe J1654−467) as representative for the outflow and place two control regions (shown by the white lines) that together cover the

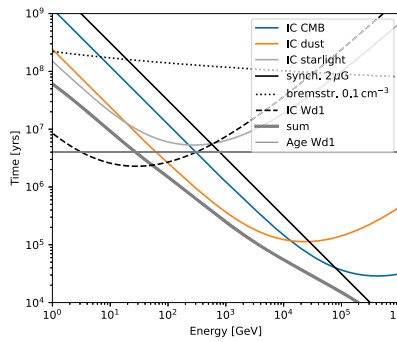
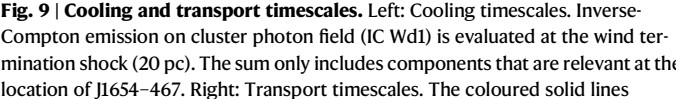
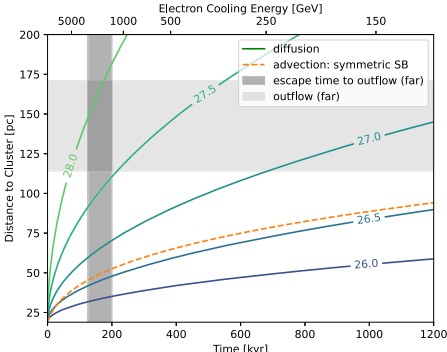

**Fig. 9 | Cooling and transport timescales.** Left: Cooling timescales. Inverse-Compton emission on cluster photon field (IC Wd1) is evaluated at the wind termination shock (20 pc). The sum only includes components that are relevant at the location of J1654−467. Right: Transport timescales. The coloured solid lines

indicate the transport for a range of diffusion coefficients, $D$, and are labelled with $\log_{10}(D/\mathrm{cm}^2\,\mathrm{s}^{-1})$. The orange dashed line indicates advection in a spherically symmetric superbubble (SB). The upper abscissa shows the energy of electrons that cool within the time given on the lower abscissa. For further details see the text.

same solid angle. For the first velocity interval ([−63, −50] km s⁻¹, left panel), the average column density in the outflow region is $\approx 5.2 \times 10^{20}\,\mathrm{cm}^{-2}$, while it is $\approx 6.7 \times 10^{20}\,\mathrm{cm}^{-2}$ in the control regions. Hence, the outflow region is less dense by about $1.5 \times 10^{20}\,\mathrm{cm}^{-2}$. Assuming an extent of the cavity along the line of sight of 70 pc, this would correspond to about 0.7 atoms cm⁻³ excavated by the outflow. Performing the same calculation for the second velocity interval ([−50, −37] km s⁻¹, right panel), we obtain a difference in density of $\approx 0.7 \times 10^{20}\,\mathrm{cm}^{-2}$, or about 0.3 atoms cm⁻³ excavated. Thus, we conclude that the outflow region is less dense by about 0.3–0.7 atoms cm⁻³ compared to its surroundings. We note that the larger difference in the first velocity interval—which corresponds to larger distances from the Earth—may indicate that the tentative outflow is not directed completely perpendicular to the line of sight, but exhibits a component directed away from us.

## Modelling details

We use the software package GAMERA[44,45] to compute the time evolution of a continuously injected particle spectrum and compute the resulting gamma-ray emission. The total and far outflow spectra (see Fig. 4) are described by two independent one-zone models. We consider inverse-Compton scattering on the CMB ($T = 2.7\,\mathrm{K}$, $U_{\mathrm{CMB}} = 0.26\,\mathrm{eV}\,\mathrm{cm}^{-3}$) and diffuse starlight and dust-scattered starlight[63]. The electron injection spectrum is $\mathrm{d}N/(\mathrm{d}E\,\mathrm{d}t) = \eta\,(E/E_0)^{-2.25}\exp(-E/170\,\mathrm{TeV})$, where the cutoff is set by the balance of acceleration and loss timescale and $\eta$ is an efficiency parameter corresponding to the fraction of star cluster wind power transferred to electrons in the acceleration process, $L_{\mathrm{inj}} = \eta L_{\mathrm{w}}$. We use $\eta = 0.7\,\%$ for $L_{\mathrm{w}} = 10^{39}\,\mathrm{erg}\,\mathrm{s}^{-1}$ above 0.01 GeV for the model shown in Fig. 4. For further details on the modelling, justification for the choice of parameters, and a discussion of the hadronic scenario see Härer et al.[41].

The left-hand plot of Fig. 9 shows the timescales for competing electron cooling mechanisms. The photon field of the massive stars in Westerlund 1 is assumed to have an effective temperature of $T_{\mathrm{eff}} = 40{,}000\,\mathrm{K}$. In the *total model*, the cluster photon field is evaluated at the cluster wind termination shock (20 pc), where the energy density is $U_{\mathrm{ph}} = 42\,\mathrm{eV}\,\mathrm{cm}^{-3}$. Since $U_{\mathrm{ph}} \propto R^{-2}$, the cluster photon field is negligible in the nascent outflow and is therefore also negligible in the *far region model*. The right-hand plot of Fig. 9 shows transport timescales for diffusion and advection inside the Westerlund 1 superbubble. We apply the same assumptions as in Härer et al.[41], for both diffusion and advection. Advection adopts a cluster wind velocity of 2500 km s⁻¹ and a scaling of the flow speed $\propto R^{-2}$ beyond the cluster wind termination shock. We plot also the transport times for diffusion coefficients $D = 10^{\alpha}\,\mathrm{cm}^2\,\mathrm{s}^{-1}$ with exponents in the range $\alpha = 26$–28. Adopting Kolmogorov scaling in a fully turbulent field (i.e. no strong guide field), the electron diffusion

coefficient is

$$D = 2.5 \times 10^{26}\left(\frac{E}{1\,\mathrm{GeV}}\right)^{1/3}\left(\frac{B}{2\mu\mathrm{G}}\right)^{-1/3}\left(\frac{R_{\mathrm{inj}}}{1\,\mathrm{pc}}\right)^{2/3}\mathrm{cm}^2\,\mathrm{s}^{-1}, \quad (1)$$

where $B$ is the average magnetic field strength and $R_{\mathrm{inj}}$ the turbulence injection scale. $D$ takes values of $1.2 \times 10^{27}\,\mathrm{cm}^2\,\mathrm{s}^{-1}$ at 100 GeV and $2.5 \times 10^{27}\,\mathrm{cm}^2\,\mathrm{s}^{-1}$ at 1 TeV. Applying the same approach using Kraichnan scaling suggests diffusion coefficients of $2.3 \times 10^{26}\,\mathrm{cm}^2\,\mathrm{s}^{-1}$ at 100 GeV and $7.2 \times 10^{26}\,\mathrm{cm}^2\,\mathrm{s}^{-1}$ at 1 TeV. The dark grey band in Fig. 9 marks the expected escape time to the far outflow, obtained from the *far region model* as described in the main text. The lower bound (125 kyr) corresponds to the model shown in Fig. 4. The upper bound (200 kyr) is obtained by changing the normalisation of the model for the entire region to match the *Fermi*-LAT points above 10 GeV. The canonical model for advection in the superbubble (orange dashed line) under-predicts the transport length-scale (about 40–50 pc compared to ≥110 pc).

## Data availability

The *Fermi*-LAT data and software needed for analysis are available from the *Fermi* Science Support Centre, https://fermi.gsfc.nasa.gov/ssc. All other data generated or analysed during this study are included in this published article (and its Supplementary information files), or referenced in the article itself. Source data are provided with this paper.

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

## Acknowledgements

The *Fermi*-LAT Collaboration acknowledges generous ongoing support from a number of agencies and institutes that have supported both the development and the operation of the LAT as well as scientific data analysis. These include the National Aeronautics and Space Administration and the Department of Energy in the United States, the Commissariat à l'Energie Atomique and the Centre National de la Recherche Scientifique/Institut National de Physique Nucléaire et de Physique des Particules in France, the Agenzia Spaziale Italiana and the Istituto Nazionale di Fisica Nucleare in Italy, the Ministry of Education, Culture, Sports, Science and Technology (MEXT), High Energy Accelerator Research Organization (KEK) and Japan Aerospace Exploration Agency (JAXA) in Japan, and the K. A. Wallenberg Foundation, the Swedish Research Council and the Swedish National Space Board in Sweden. Additional support for science analysis during the operations phase is gratefully acknowledged from the Istituto Nazionale di Astrofisica in Italy and the Centre National d'Études Spatiales in France. This work performed in part under DOE Contract DE-AC02-76SF00515. Marianne Lemoine-Goumard acknowledges support from the Alexander von Humboldt Foundation. The authors thank Jesús Maíz Apellániz, Ignacio Negueruela Díez, and Quentin Remy for discussions about the interstellar gas in the vicinity of Westerlund 1.

## Author contributions

M.L.-G. performed the *Fermi*-LAT data analysis. L.M. carried out the study of the ISM density and created all figures (except Fig. 9). L.H. developed the theoretical modelling and produced Fig. 9, with support by B.R. and T.V. R.B., G.P. and L.T. assisted with the *Fermi*-LAT data analysis; L.T. furthermore helped with the ISM density study. M.L.-G., L.H., L.M., J.H. and B.R. wrote the paper. All authors have provided comments to initial versions of the article.

## Funding

## Competing interests

The authors declare no competing interests.
