## [Transparent Peer Review file · Nature Communications]

A cosmic-ray loaded nascent outflow driven by a massive star cluster

Corresponding Author: Professor Marianne Lemoine-Goumard

Version 0:

Reviewer comments:

Reviewer #1

(Remarks to the Author)

The manuscript illustrates the discovery of gamma ray emission in the ~ 10 GeV range, accessible to Fermi-LAT, from a region that appears to be related to the Westerlund 1 star cluster. The SED of this region is connected with continuity to the HESS emission of the star cluster in the high energy range. The authors argue that their analysis of Fermi-LAT data shows that this is an outflow driven by the star cluster. Given the rising interest for the case of star clusters as potential cosmic ray accelerators, I think the topic is timely and of interest to a wide pool of readers.

Overall, the manuscript is well written, concise and contains appendixes that explain several technical details of the analysis and of the phenomenological interpretation.

I think the manuscript illustrates results worth being published, but I think that the part discussing scientific implications and modelling is not as clear and well justified. It contains several points which seem to me to be not properly justified. Hence I would like the authors to address the points below before I make my final decision on recommending publication.

1) In several places in the text it is alluded to the fact that the cosmic rays might play a dynamical role in the formation of the bubble around Westerlund 1 and in starting the outflow penetrating through the disc outward. All considerations in the manuscript are based on the acceleration of electrons and hence inferring the cosmic ray (mainly hadronic) energy content is a matter of speculation. But even if the argument is clarified, and I expect that the authors would just assume the same ratio of protons to electrons as inferred in the Milky Way at large, the efficiency of particle acceleration remains appreciably smaller than unity (as it should, of course), hence the main contribution to the pressure most likely comes from the gas ram pressure plus internal energy. This is, after all, the reason why the bubble around a star cluster is formed in the first place (even without cosmic rays). Why should the “new structure” revealed by this analysis simply not be due to the change of shape of the same cavity as it penetrates the region outside the disc, as due to the same processes that form the rest of the bubble? Clearly one could speculate that particles leaving the star cluster may be generating the outflow, but I did not find any argument in the manuscript to show that this may be the case and no discussion of the escape of these particles from the star cluster is present.

2) At line 160 the authors start a discussion of the modelling and claim that what they name “the total model” provides a good description of the data, as resulting from leptonic emission of electrons accelerated at the termination shock of the star cluster. This model is nothing more than what was presented by a subset of the authors in Ref. [39]. This is understandable and it would provide support to my previous statement, namely that what we are seeing is the asymmetric expansion of the bubble when its boundary crosses the disc region, while the overall emission is compatible with what one would expect in a simple model of the bubble (Ref. [39]). In this picture we are not seeing evidence of cosmic rays anomalous transport but simply the fact that they are still inside the acceleration/source region. On the other hand, at line 172, the authors present another model of what they call the “far region”, but there is not enough discussion of this model for me to assess its viability. It sounds something similar to a fit rather than a model and not enough information is provided for me to understand whether this is viable as an explanation of what the authors observed. I hope that the authors may decide to add some discussion of what has been done here and perhaps discuss why it should (or if it should) be preferred to the total model.

3) On Line 183 the authors write “on the scales of interest, diffusive transport is certainly be both anisotropic and non uniform”. This conclusion is based on the previous statement that “The timescale of transport to the “far” region is much

shorter than expected for advection in a spherically symmetric super-bubble". Now, the fact that the bubble is penetrating outside the disc obviously leads to the conclusion that the bubble cannot be spherically symmetric, so no surprise there. But the fact that this implies that diffusive transport must be both anisotropic and non uniform seems unjustified. This is clearly true for the "anisotropic" part of the statement, since the authors never advocate for specific directions of local magnetic fields. The non uniformity of the diffusion process is again related to the ambiguity of the manuscript in terms of what is inside and outside the bubble. All descriptions of the gamma ray emission from the region of Westerlund (including Ref. [39]) require a diffusion coefficient in the bubble that is much smaller than usually inferred for the Galaxy at large, but this is no surprise since that is the source region. For the same reason, nobody would be claiming that the transport in the Galaxy is non uniform because inside supernova remnants diffusivity is required to be suppressed. To be clear: I am not claiming here that transport in the Galaxy is isotropic and/or uniform, I am just saying that no such conclusion, one way or another, can be reached based on the observations illustrated in this manuscript. If the authors think differently, then a much better case need to be made that the "far region" is due to particles other than the ones inside the super-bubble.

4) At line 203 the authors claim that their estimate of the cosmic ray energy density in the region is much higher than the in the general interstellar medium and this implies that cosmic rays play a role in driving the outflow. Again, this is a misconception: it is rather intuitively clear that if one looks inside or around a source (whatsoever the source is) the energy density is going to be higher than average. In the specific case discussed here, the bubble is excavated by the collective stellar wind (even without cosmic rays) and both the energy density of accelerated particles and the diffusivity inside the bubble are likely to be anomalous if compared with the interstellar medium. This does not mean that the outflow is being driven by the cosmic rays. Again, if the authors want to make that point, then they should build a better case for cosmic rays leaving the region. This is probably not the case since the "total model" that they presented provides an excellent description of the data, even if all the emission comes from inside the bubble. If the whole point is that the bubble becomes asymmetric and that this observation shows it, then I think that that point can be made more clearly and without invoking the dynamical role of accelerated particles.

Reviewer #2

(Remarks to the Author)

The paper reports the discovery of gamma-ray emission linking the most massive young stellar cluster in the Milky Way to its large-scale outflow. This detection of an early-stage breakout outflow from the cluster, transporting cosmic rays (CRs) with an energy density significantly exceeding the Galactic mean, provides important insight into the role of superbubbles in driving galactic winds and polluting the circumgalactic medium with CRs. This is a novel and significant result. We recommend that the paper be published after the authors address the following suggestions aimed at improving the manuscript.

The authors present new Fermi-LAT (~GeV) data alongside previous HESS (~TeV) and GASS (HI 21cm) observations. They find that the Fermi observations show a significant GeV source below (away from the disk) Westerlund 1. This source is spatially coincident with a region of low HI column density. Additionally, when splitting the spectrum between "near" (close to the disk, close to Westerlund 1) and "far" (away from the disk, further from Westerlund 1), the "far" GeV spectrum shows signs of "aging." This has significant implications in galactic feedback, galactic outflows, and CR feedback. CR escape into the halo may be an inhomogeneous process, allowing for large amounts of CRs to be directly "handed over" to the halo by star cluster-driven outflows, without significant ISM losses. Additionally, the high value of the inferred CR energy density within the HI cavity suggests that CRs may potentially be dynamically significant in star cluster-driven outflows.

The paper would benefit from a slightly expanded discussion of the caveats of the superbubble model and interpretation of the results. Specifically, the authors state that "the estimated CR energy density is more than an order of magnitude beyond that of the general interstellar medium, making it likely that CRs play a significant role in driving the outflow." However, the ability of CRs to accelerate an outflow depends on the phase of gas they are interacting with (Armillotta et al. (2024), already cited; Sike et al. (2025), submitted to ApJ, <https://arxiv.org/abs/2410.06988>; Kjellgren et al. (2025), submitted to A&A, <https://arxiv.org/abs/2502.02635>). This is an especially important consideration if the outflow is being attributed to a superbubble breakout, because this gas would presumably be hot, and CRs are expected to not contribute to the acceleration of a hot outflow as argued in the above papers. A brief discussion of this limitation and its implications for the interpretation would strengthen the paper.

The authors suggest that "the timescale of transport to the "far" region is much shorter than expected for advection in a spherically symmetric superbubble." (near line 182). Fig. A9 suggests that, when quantified in terms of the diffusion coefficient, the diffusion coefficient increases from $\sim 10^{26.5}$ to $\sim 10^{27.5}$ (based on the location of the intersection of the horizontal line and the gray-shaded vertical region). Could the authors explicitly quantify (near line 182) the magnitude of the increase in CR transport speed? (subject to the caveats discussed below).

The above conclusion regarding the speed of CR transport is based on a model of the cluster wind that assumes a specific normalization (2500 km s^{-1}) and radial dependence of the outflow velocity (propto r^{-2}). Alternatively, based on the spectral aging (125 kyr) and the spatial scale ($\sim 70 \text{ pc}$), one could derive an approximate (constant) expansion velocity of $\sim 550 \text{ km/s}$. This magnitude of velocity could be explained in an advection scenario within the superbubble breakout model. This would support the idea that the outflow is hot, the CRs are primarily advected, and, therefore, that (i) the CRs do not necessarily need to significantly contribute to the acceleration of the wind, and that (ii) the speed of CR transport does not have to be elevated beyond what is expected in superbubble outflows. Therefore, the conclusion may be sensitive to the details of the modeling of the superbubble outflow.

Nevertheless, eventual escape of CRs from the cavity followed by their recoupling to the ISM may substantially increase the ability of CRs to drive winds. Overall, this does not detract from the importance of the main result: that a superbubble breakout outflow is present and “loaded” with CRs characterized by high energy density, which may become dynamically significant.

Figures 1 and 2: Hadronic gamma-ray emission from hot, underdense regions—such as the cavity coincident with the peak Fermi emission—is expected to be suppressed relative to emission from denser gas. However, the authors argue that the CR proton energy density is significantly enhanced in this cavity relative to typical ISM values. This raises the question of whether interactions between these CRs and the denser HI “walls” of the cavity (as seen in Fig. 2) might produce a localized enhancement of gamma-ray emission. Is the apparent lack of such a feature in Fig. 1 consistent with expectations for hadronic emission?

Fig. 3: The authors assume magnetic field of $2\mu\text{G}$. Caption to Fig. A9 in the Appendix suggests that synchrotron losses are negligible for this strength of the magnetic field “at the location of J1654-467.” It may be helpful to acknowledge that the synchrotron contribution to the spectrum shown in Fig. 3 is negligible (somewhere near line 166). Is any radio signature expected for this magnetic field strength?

Fig. A9, right panel: What is the definition of “Kol. Diff”? When discussing diffusion in the context of this figure, the authors state that “For diffusion, we take the turbulence injection scale to be 1pc and assume a magnetic field of $2\mu\text{G}$.” Could the authors provide more details regarding how diffusion was modeled?

Reviewer #3

(Remarks to the Author)
Dear authors,

I am sorry for the delay in the report of “A cosmic-ray loaded outflow driven by a massive star cluster”:

This paper presents a novel observational study of a cosmic-ray (CR) loaded outflow from the young massive star cluster Westerlund 1. The authors report the discovery of a structured gamma-ray emission feature (J1654–467), extending approximately 150 parsecs from the cluster, which spatially coincides with a low-density cavity in atomic hydrogen. The spectral energy distribution, spanning from GeV (Fermi-LAT) to TeV (H.E.S.S.) energies, connects smoothly across this structure, indicating a common origin. The most noteworthy result is the identification of a large-scale outflow that not only carries cosmic rays but may also be significantly shaped by their pressure. This constitutes some of the first direct observational evidence for a CR-loaded outflow associated with a resolved superbubble, validating theoretical expectations long proposed in the literature.

The paper covers the relevant physics and discusses the emission analysis in detail. Overall, the paper is of high interest to the community and deserves to be published. In particular, it has implications beyond high-energy astrophysics, addressing assumptions in galaxy evolution models where CRs are thought to help drive winds. While CR feedback is often included in simulations, direct observational evidence has been scarce. By resolving a CR-rich outflow from a specific cluster, the study provides key empirical support.

However, one part of the discussion is not clearly outlined. The authors suggest that CRs can drive galactic winds and then link this to the outflows from Westerlund 1, aiming to connect small and large scales. However, these two outflows are not directly related, and the line of argument is more subtle. Galactic outflows driven by CRs (in simulations) emerge over Gyr timescales out to tens of kiloparsec heights as a result of sustained CR pressure gradients that exert a net outward force. These CRs accumulate from many individual acceleration sites and events—numerous wind bubbles and SN shocks. The effect of galactic outflows is therefore not a question of whether CRs can locally drive outflows or where they in detail originate from. On these long timescales, the relevant question is more about whether they survive long enough to fill the ISM with high energy densities and impact the galaxy-wide ISM coherently.

While the details of the cluster and the associated CR outflow are well explained, the broader argument—that CRs are dynamically important on galactic scales—would benefit from a discussion of losses and/or transport timescales. Although complex modeling may be beyond the scope of the paper, an approximate estimate of the loss timescale would help substantiate the claim that CRs play a dominant role, rather than being a by-product. I suggest either adding a discussion of CR cooling/losses for protons at the relevant energy to drive winds (1-few GeV/c)—since they may not be negligible—or restructuring the argument so that the relevance to galactic scales is framed as conditional on losses not dominating. The authors discuss the electron time scales and refer to Häger et al. 2023, but this does not allow for a connection from local to global scales.

The conclusions regarding the observations are well supported. The authors present strong evidence for CR acceleration and local escape from Westerlund 1, and they associate the gamma-ray emission with a large-scale flow (local outflow). The suggestion that CRs influence this flow is plausible given their high inferred energy density, though direct proof is lacking. The paper appropriately presents this influence as a possibility, not a definitive conclusion.

The data interpretation is robust. The HI cavity identification is convincing, supported by multiple velocity slices. The modeling—using the GAMERA code—is appropriate and clearly explained. The authors also acknowledge the limitations of their single-zone model and emphasize the need for more detailed simulations.

The paper provides sufficient methodological detail for replication by expert readers. While full reproduction may require access to specific spatial templates and Fermi-LAT background models, the overall transparency is consistent with current standards in the field.

Reviewer #4

(Remarks to the Author)

Version 1:

Reviewer comments:

Reviewer #1

(Remarks to the Author)

All my previous concerns were properly addressed, hence I can now recommend the manuscript for publication.

Reviewer #2

(Remarks to the Author)

Thank you for submitting the revised version of your manuscript and for providing detailed responses to my comments. The revised paper and your answers thoroughly address the concerns I raised. I am satisfied with the revisions and recommend that the paper be accepted for publication.

Reviewer #3

(Remarks to the Author)

Dear Editor,

I thank the authors for their detailed clarifications and the additional explanations in the revised manuscript. The streamlined text, the newly added sketch, and the sharpened distinction between halo-driven Galactic outflows and the nascent outflow inferred from the gamma-ray maps address my earlier concerns. These changes make the manuscript clearer and help to avoid potential misunderstandings regarding the scope of the conclusions.

I am satisfied that my questions have been addressed, and I recommend the paper for publication in its present form.

Reviewer #4

(Remarks to the Author)

We thank the reviewers for their assessment of our manuscript and for providing constructive suggestions for improvements. Below we provide answers to the specific comments. Changes to the manuscript are marked in bold orange.

REVIEWER COMMENTS

Reviewer #1 (Remarks to the Author):

The manuscript illustrates the discovery of gamma ray emission in the ~ 10 GeV range, accessible to Fermi-LAT, from a region that appears to be related to the Westerlund 1 star cluster. The SED of this region is connected with continuity to the HESS emission of the star cluster in the high energy range. The authors argue that their analysis of Fermi-LAT data shows that this is an outflow driven by the star cluster. Given the rising interest for the case of star clusters as potential cosmic ray accelerators, I think the topic is timely and of interest to a wide pool of readers.

Overall, the manuscript is well written, concise and contains appendixes that explain several technical details of the analysis and of the phenomenological interpretation.

I think the manuscript illustrates results worth being published, but I think that the part discussing scientific implications and modelling is not as clear and well justified. It contains several points which seem to me to be not properly justified. Hence I would like the authors to address the points below before I make my final decision on recommending publication.

1) In several places in the text it is alluded to the fact that the cosmic rays might play a dynamical role in the formation of the bubble around Westerlund 1 and in starting the outflow penetrating through the disc outward. All considerations in the manuscript are based on the acceleration of electrons and hence inferring the cosmic ray (mainly hadronic) energy content is a matter of speculation. But even if the argument is clarified, and I expect that the authors would just assume the same ratio of protons to electrons as inferred in the Milky Way at large, the efficiency of particle acceleration remains appreciably smaller than unity (as it should, of course), hence the main contribution to the pressure most likely comes from the gas ram pressure plus internal energy. This is, after all, the reason why the bubble around a star cluster is formed in the first place (even without cosmic rays). Why

should the “new structure” revealed by this analysis simply not be due to the change of shape of the same cavity as it penetrates the region outside the disc, as due to the same processes that form the rest of the bubble? Clearly one could speculate that particles leaving the star cluster may be generating the outflow, but I did not find any argument in the manuscript to show that this may be the case and no discussion of the escape of these particles from the star cluster is present.

We thank the reviewer for providing positive feedback, and their suggestions to improve the manuscript. We agree that some of the wording on the dynamic significance of the inferred CR component was too speculative and/or insufficiently clear. We have softened the relevant statements in the abstract and conclusion section.

With regards to the CR efficiency, the referee is correct that we are making a (we believe reasonable) assumption regarding the proton content based on diffuse shock acceleration theory, and of course it should not exceed the gas pressure in the interior bubble. However, as the outflow develops, and the cold shell disrupts, material at high latitudes will inevitably cool. Here, the more mobile CRs may diffuse to larger distances, where they may begin to dominate the pressure and play a more substantial role. The open question is how and in what quantity they eventually reach the halo, but as the referee points out this is hard to probe with the existing data. We have re-phrased in multiple places, including changing the title, to emphasise that we are currently witnessing a *nascent* outflow. Our hope is that our findings will provide a valuable constraint on future modelling of CR/gas transport into the halo, and that future observatories with improved sensitivity will bring even stronger constraints.

2) At line 160 the authors start a discussion of the modelling and claim that what they name “the total model” provides a good description of the data, as resulting from leptonic emission of electrons accelerated at the termination shock of the star cluster. This model is nothing more than what was presented by a subset of the authors in Ref. [39]. This is understandable and it would provide support to my previous statement, namely that what we are seeing is the asymmetric expansion of the bubble when its boundary crosses the disc region, while the overall emission is compatible with what one would expect in a simple model of the bubble (Ref. [39]). In this picture we are not seeing evidence of cosmic rays anomalous transport but simply the fact that they are still inside the acceleration/source region. On the other hand, at line 172, the authors present another model of what

they call the “far region”, but there is not enough discussion of this model for me to assess its viability. It sounds something similar to a fit rather than a model and not enough information is provided for me to understand whether this is viable as an explanation of what the authors observed. I hope that the authors may decide to add some discussion of what has been done here and perhaps discuss why it should (or if it should) be preferred to the total model.

We agree and partially addressed this comment in our response to point 1). Regarding the latter part of the comment, it was not our intention to present the “far region” model as an alternative model to the “total” model. Instead, we wanted to demonstrate that the spectrum of the emission in the “far” region is consistent with transport of cosmic rays accelerated at the cluster wind termination shock into that region, i.e. in the framework of the “total” model. We have now modified the text to better motivate the “far region” model in the manuscript. We also revised the Appendix and improved Figure A9 to make the relevant transport requirements easier to visualise/understand.

3) On Line 183 the authors write “on the scales of interest, diffusive transport is certainly be both anisotropic and non uniform”. This conclusion is based on the previous statement that “The timescale of transport to the “far” region is much shorter than expected for advection in a spherically symmetric super-bubble”. Now, the fact that the bubble is penetrating outside the disc obviously leads to the conclusion that the bubble cannot be spherically symmetric, so no surprise there. But the fact that this implies that diffusive transport must be both anisotropic and non uniform seems unjustified. This is clearly true for the “anisotropic” part of the statement, since the authors never advocate for specific directions of local magnetic fields. The non uniformity of the diffusion process is again related to the ambiguity of the manuscript in terms of what is inside and outside the bubble. All descriptions of the gamma ray emission from the region of Westerlund (including Ref. [39]) require a diffusion coefficient in the bubble that is much smaller than usually inferred for the Galaxy at large, but this is no surprise since that is the source region. For the same reason, nobody would be claiming that the transport in the Galaxy is non uniform because inside supernova remnants diffusivity is required to be suppressed. To be clear: I am not claiming here that transport in the Galaxy is isotropic and/or uniform, I am just saying that no such conclusion, one way or another, can be reached based on the observations illustrated in this manuscript. If the authors think

differently, then a much better case need to be made that the “far region” is due to particles other than the ones inside the super-bubble.

These statements were intended to highlight that the situation is more complex than what is captured by our rather simplistic model, calling for more detailed modelling efforts in the future. Our initial thinking was that the asymmetric outflow is likely to enforce a magnetic topology that affects the transport. But such arguments are not essential for the current investigation, and to avoid any misunderstanding/over-claims we have simply removed the sentence in question.

On the general aspects of presentation of our interpretation with regards to what is inside versus outside the bubble, we agree that the original draft was not sufficiently clear and have made several text changes (see also response to comment 4 below) and added a new sketch that hopefully helps to clarify the scenario that we have in mind.

4) At line 203 the authors claim that their estimate of the cosmic ray energy density in the region is much higher than the in the general interstellar medium and this implies that cosmic rays play a role in driving the outflow. Again, this is a misconception: it is rather intuitively clear that if one looks inside or around a source (whatsoever the source is) the energy density is going to be higher than average. In the specific case discussed here, the bubble is excavated by the collective stellar wind (even without cosmic rays) and both the energy density of accelerated particles and the diffusivity inside the bubble are likely to be anomalous if compared with the interstellar medium. This does not mean that the outflow is being driven by the cosmic rays. Again, if the authors want to make that point, then they should build a better case for cosmic rays leaving the region. This is probably not the case since the “total model” that they presented provides an excellent description of the data, even if all the emission comes from inside the bubble. If the whole point is that the bubble becomes asymmetric and that this observation shows it, then I think that that point can be made more clearly and without invoking the dynamical role of accelerated particles.

We agree again with the referee, and we do not wish to claim that we have evidence that the outflow is mainly driven by CRs. We have rephrased where appropriate to avoid misunderstanding. Nevertheless, we argue that it is fair to make a distinction between the region immediately surrounding the star cluster (i.e. close to the acceleration site near the wind termination shock, the region where

the highest energy electrons appear to be radiating) and the nascent outflow region. CRs in the latter region are clearly on their way out of the Galactic plane and will in time lose their connection to the region dominated by the cluster wind. We see the (intriguing) current situation of Wd1 as intermediate between the 10s of parsec scales of wind-driven bubbles and the kpc-scales of the large-scale Galactic outflows. We hope that the revised statements are acceptable to the referee.

Reviewer #2 (Remarks to the Author):

The paper reports the discovery of gamma-ray emission linking the most massive young stellar cluster in the Milky Way to its large-scale outflow. This detection of an early-stage breakout outflow from the cluster, transporting cosmic rays (CRs) with an energy density significantly exceeding the Galactic mean, provides important insight into the role of superbubbles in driving galactic winds and polluting the circumgalactic medium with CRs. This is a novel and significant result. We recommend that the paper be published after the authors address the following suggestions aimed at improving the manuscript.

The authors present new Fermi-LAT (\sim GeV) data alongside previous HESS (\sim TeV) and GASS (HI 21cm) observations. They find that the Fermi observations show a significant GeV source below (away from the disk) Westerlund 1. This source is spatially coincident with a region of low HI column density. Additionally, when splitting the spectrum between "near" (close to the disk, close to Westerlund 1) and "far" (away from the disk, further from Westerlund 1), the "far" GeV spectrum shows signs of "aging." This has significant implications in galactic feedback, galactic outflows, and CR feedback. CR escape into the halo may be an inhomogeneous process, allowing for large amounts of CRs to be directly "handed over" to the halo by star cluster-driven outflows, without significant ISM losses. Additionally, the high value of the inferred CR energy density within the HI cavity suggests that CRs may potentially be dynamically significant in star cluster-driven outflows.

The paper would benefit from a slightly expanded discussion of the caveats of the superbubble model and interpretation of the results. Specifically, the authors state that "the estimated CR energy density is more than an order of magnitude beyond that of the general interstellar medium, making it likely that CRs play a significant role in driving the outflow." However, the ability of CRs to accelerate an outflow depends on the phase of gas they are interacting

with (Armillotta et al. (2024), already cited; Sike et al. (2025), submitted to ApJ, <https://arxiv.org/abs/2410.06988>; Kjellgren et al. (2025), submitted to A&A, <https://arxiv.org/abs/2502.02635>). This is an especially important consideration if the outflow is being attributed to a superbubble breakout, because this gas would presumably be hot, and CRs are expected to not contribute to the acceleration of a hot outflow as argued in the above papers. A brief discussion of this limitation and its implications for the interpretation would strengthen the paper.

We thank the referee for raising this point and for the specific suggestions. We have included these recent works, as they indeed highlight the influence that CRs play in Galactic outflows, and are in fact an essential ingredient to drive the winds beyond the multi-kpc scale. Very few of the multiphase kpc scale simulations to date have included the mechanical feedback from the cluster winds (though see Rathjen et al 2021, now cited), which we argue is the primary driver for the observed emission. We believe our findings will stimulate further exploration with these multi-scale simulations.

The authors suggest that “the timescale of transport to the “far” region is much shorter than expected for advection in a spherically symmetric superbubble.” (near line 182). Fig. A9 suggests that, when quantified in terms of the diffusion coefficient, the diffusion coefficient increases from $\sim 10^{26.5}$ to $\sim 10^{27.5}$ (based on the location of the intersection of the horizontal line and the gray-shaded vertical region). Could the authors explicitly quantify (near line 182) the magnitude of the increase in CR transport speed? (subject to the caveats discussed below).

On re-reading this section we found that we were not sufficiently clear in the Figure nor the text. We have simplified the figure and (we believe) improved the text on this aspect, adding an expression for the diffusion coefficient required.

The above conclusion regarding the speed of CR transport is based on a model of the cluster wind that assumes a specific normalization (2500 km s^{-1}) and radial dependence of the outflow velocity ($\propto r^{-2}$). Alternatively, based on the spectral aging (125 kyr) and the spatial scale ($\sim 70 \text{ pc}$), one could derive an approximate (constant) expansion velocity of $\sim 550 \text{ km/s}$. This magnitude of velocity could be explained in an advection scenario within the superbubble breakout model. This would support the idea that the outflow is hot, the CRs are

primarily advected, and, therefore, that (i) the CRs do not necessarily need to significantly contribute to the acceleration of the wind, and that (ii) the speed of CR transport does not have to be elevated beyond what is expected in superbubble outflows. Therefore, the conclusion may be sensitive to the details of the modeling of the superbubble outflow.

We agree that the conclusions on the transport are highly sensitive to the velocity profile assumed for the nascent outflow. Assuming constant speed, the required velocity is ~500-1000 km/s depending on the scales considered (125-200 kyr and 90-120 pc for the distance of the J1654-467 from the wind termination shock). These values seem rather high, considering that the free-wind speed is reduced by a factor four at the shock. In addition, within ~20 pc from the termination shock, the geometry is approximately spherical and the flow is expected to expand and slow down approximately as r^{-2} . It seems therefore likely that diffusion plays a role for CR transport in the nascent outflow. Further conclusions cannot be drawn at the current stage of modelling. Therefore we present only the spherically symmetric model for advection and curves for constant diffusion coefficients in the paper.

Nevertheless, eventual escape of CRs from the cavity followed by their recoupling to the ISM may substantially increase the ability of CRs to drive winds. Overall, this does not detract from the importance of the main result: that a superbubble breakout outflow is present and "loaded" with CRs characterized by high energy density, which may become dynamically significant.

We thank the reviewer for this favourable assessment of our result. We have endeavoured to sharpen this aspect throughout the revised manuscript.

Figures 1 and 2: Hadronic gamma-ray emission from hot, underdense regions—such as the cavity coincident with the peak Fermi emission—is expected to be suppressed relative to emission from denser gas. However, the authors argue that the CR proton energy density is significantly enhanced in this cavity relative to typical ISM values. This raises the question of whether interactions between these CRs and the denser HI "walls" of the cavity (as seen in Fig. 2) might produce a localized enhancement of gamma-ray emission. Is the apparent lack of such a feature in Fig. 1 consistent with expectations for hadronic emission?

The reviewer is correct in general, but we argue that it is not expected that such a feature is detected in our *Fermi*-LAT analysis. First, the density of the ISM in this region below the Galactic plane is quite low overall, so that even in the swept-up “walls” the density will be modest. Second, the resolution of *Fermi*-LAT is quite limited, and part of the emission coming from the “walls” is likely to be absorbed in the background models that have been adjusted to the data. Disentangling this component will be an extremely challenging task. Hence we cannot claim that no such emission is present, but the lack thereof in (e.g.) Fig. 1 is not inconsistent with our picture.

Fig. 3: The authors assume magnetic field of $2\ \mu\text{G}$. Caption to Fig. A9 in the Appendix suggests that synchrotron losses are negligible for this strength of the magnetic field “at the location of J1654-467.” It may be helpful to acknowledge that the synchrotron contribution to the spectrum shown in Fig. 3 is negligible (somewhere near line 166). Is any radio signature expected for this magnetic field strength?

We added statements to the paper saying that indeed the contribution of synchrotron radiation at GeV and TeV energies is negligible, and also that the level of synchrotron emission predicted by the model is well below upper limits derived from radio continuum maps. A corresponding appendix with a figure demonstrating this has been added.

Fig. A9, right panel: What is the definition of “Kol. Diff”? When discussing diffusion in the context of this figure, the authors state that “For diffusion, we take the turbulence injection scale to be 1pc and assume a magnetic field of $2\ \mu\text{G}$.” Could the authors provide more details regarding how diffusion was modeled?

We have removed the lines labelled “Kol. Diff.” from the plot for clarity. We now provide an equation for the diffusion coefficient in the Kolmogorov case in the text and give values at two different energies. We hope that this clarifies the reviewer’s question.

Reviewer #3 (Remarks to the Author):

Dear authors,

I am sorry for the delay in the report of "A cosmic-ray loaded outflow driven by a massive star cluster":

This paper presents a novel observational study of a cosmic-ray (CR) loaded outflow from the young massive star cluster Westerlund 1. The authors report the discovery of a structured gamma-ray emission feature (J1654–467), extending approximately 150 parsecs from the cluster, which spatially coincides with a low-density cavity in atomic hydrogen. The spectral energy distribution, spanning from GeV (Fermi-LAT) to TeV (H.E.S.S.) energies, connects smoothly across this structure, indicating a common origin. The most noteworthy result is the identification of a large-scale outflow that not only carries cosmic rays but may also be significantly shaped by their pressure. This constitutes some of the first direct observational evidence for a CR-loaded outflow associated with a resolved superbubble, validating theoretical expectations long proposed in the literature.

The paper covers the relevant physics and discusses the emission analysis in detail. Overall, the paper is of high interest to the community and deserves to be published. In particular, it has implications beyond high-energy astrophysics, addressing assumptions in galaxy evolution models where CRs are thought to help drive winds. While CR feedback is often included in simulations, direct observational evidence has been scarce. By resolving a CR-rich outflow from a specific cluster, the study provides key empirical support.

However, one part of the discussion is not clearly outlined. The authors suggest that CRs can drive galactic winds and then link this to the outflows from Westerlund 1, aiming to connect small and large scales. However, these two outflows are not directly related, and the line of argument is more subtle. Galactic outflows driven by CRs (in simulations) emerge over Gyr timescales out to tens of kiloparsec heights as a result of sustained CR pressure gradients that exert a net outward force. These CRs accumulate from many individual acceleration sites and events—numerous wind bubbles and SN shocks. The effect of galactic outflows is therefore not a question of whether CRs can locally drive outflows or where they in detail originate from. On these long timescales, the relevant question is more about whether they survive long enough to fill the ISM with high energy densities and impact the galaxy-wide ISM coherently.

We do not aim to draw conclusions about kpc-scale outflows from the results obtained in our work, and we have streamlined in particular the introduction section to avoid giving this impression. Furthermore,

we have added a sketch that we hope clarifies the scenario that we have in mind. Nevertheless, in our view a better understanding of intermediate-scale outflows such as the one we observe is an important ingredient to learning about galaxy-scale outflows (hence the line of argument in the introduction).

While the details of the cluster and the associated CR outflow are well explained, the broader argument—that CRs are dynamically important on galactic scales—would benefit from a discussion of losses and/or transport timescales. Although complex modeling may be beyond the scope of the paper, an approximate estimate of the loss timescale would help substantiate the claim that CRs play a dominant role, rather than being a by-product. I suggest either adding a discussion of CR cooling/losses for protons at the relevant energy to drive winds (1-few GeV/c)—since they may not be negligible—or restructuring the argument so that the relevance to galactic scales is framed as conditional on losses not dominating. The authors discuss the electron time scales and refer to Härer et al. 2023, but this does not allow for a connection from local to global scales.

We thank the referee for this comment. In the revised draft, we have sharpened the distinction between Galactic outflows which are driven in the halo, and the nascent outflow we infer from the gamma-ray maps. We do not claim that CRs play the dominant role in this nascent outflow. But the loss timescales for any hadronic component in such low density environments is much longer than the current lifetime of the Westerlund 1 system.

The conclusions regarding the observations are well supported. The authors present strong evidence for CR acceleration and local escape from Westerlund 1, and they associate the gamma-ray emission with a large-scale flow (local outflow). The suggestion that CRs influence this flow is plausible given their high inferred energy density, though direct proof is lacking. The paper appropriately presents this influence as a possibility, not a definitive conclusion.

The data interpretation is robust. The HI cavity identification is convincing, supported by multiple velocity slices. The modeling—using the GAMERA code—is appropriate and clearly explained. The authors also acknowledge the limitations of their single-zone model and emphasize the need for more detailed simulations.

The paper provides sufficient methodological detail for replication by expert readers. While full reproduction may require access to specific spatial templates and Fermi-LAT background models, the overall transparency is consistent with current standards in the field.

We thank the referee for these additional supportive remarks.

Reviewer #4 (Remarks to the Author):
